# Capacitive Effect and Electromagnetic Coupling on Manganin Gauge Limiting the Bandwidth for Pressure Measurements under Shock Conditions

**DOI:** 10.3390/s23146583

**Published:** 2023-07-21

**Authors:** Antony Coustou, Alexandre Lefrançois, Patrick Pons, Yohan Barbarin

**Affiliations:** 1CNRS-LAAS, Toulouse University, 7 avenue du Colonel Roche, F-31031 Toulouse, France; ppons@laas.fr; 2CEA-DAM, GRAMAT, F-46500 Gramat, France; alexandre.lefrancois@cea.fr (A.L.); yohan.barbarin@cea.fr (Y.B.)

**Keywords:** piezoresistivity, shock properties, manganin gauge, metrology, shock pressure, capacitive effect, electromagnetic coupling

## Abstract

In this study, we investigated the capacitive effect and the electromagnetic coupling on the measurement chain induced by impact experiments with a gas gun or powder gun. Reduced bandwidth and noise were noticed on experimental signals. Rogowski coil measurements were added on the cables to characterize the electromagnetic coupling. The perturbation currents on the cables were quantified depending on the configuration. The gauge, the transmission line and the conditioning system were modeled. The calculations reproduced the electrical wave arrival time, the transmission line transfer impedance and the conditioning system transfer impedance; and the bandwidth limitation has been displayed. A capacitive effect with the piezoresistive manganin gauge embedded into the sample was identified, depending on the experimental setup.

## 1. Introduction

Pressure measurements in materials under shock loading are essential to calibrate or validate the Hugoniot equations of state or reactive processes induced in energetic materials, such as the shock-to-detonation behavior in high explosives.

The piezoresistive technique has been applied for decades with a proportional relationship between the variation of the electrical resistance, the stress and the strain (1) [1,2,3]. For a plane shock experiment, associated with a controlled longitudinal strain, the variation of the resistance is proportional to the longitudinal stress or transverse stress, depending on the position of the gauge [4,5].
(1)$$\frac{\Delta R}{R_{0}}=a\cdot ∆\sigma +k\cdot (\varepsilon +\varepsilon_{d}\left(T\right))$$
where *R* is the gauge resistance, *R*_0_ the initial resistance, *a* the piezoresistive coefficient, *σ* the stress, *ε* the strain, *k* the gauge factor and *ε_d_* the dilatation strain with the initial temperature *T*, which is different from the calibrated temperature.

High-impedance (25, 50 or 120 ohm) and low-impedance (tens of milliohm) manganin gauges are often used within the target. These are strain gauges applied for shock experiments, called stress gauges, or even pressure gauges. The piezoresistive coefficients could vary depending on the gauge thickness and on the alloy type. The pressure uncertainty needs to be evaluated, considering different sources. This could lead to high uncertainty on the gauge sensitivity. The conditioning system is usually a resistive Wheatstone bridge for high-impedance gauge with voltage, current power supply or a direct current power supply for both types of gauges [6,7,8]. The uncertainties on the resistive bridge calibrations and of the piezoresistive coefficients are identified, respectively, with verified ohmmeter and digitizer uncertainties and with the shock conditions. Present measurement uncertainty has been evaluated between 5 and 10% for high-impedance gauges [9].

Very often, rise time issues, noise and signal oscillations have been observed over the years [10,11,12].

The main research has been focused on the piezoresistive gauge calibration for low-impedance gauges [13,14,15,16,17] and high-impedance gauges [14,15,18,19], on the conditioning device development [6,12,13] and on gauge issues, such as hysteresis [14,20], piezoresistive behavior during the release [21], piezoresistive behavior in the elastic domain and in the plastic domain of the gauge [22,23]. The signal rise time of the gauge, limited to its thickness, has been demonstrated [24]. Noise disturbances have been observed and mitigated previously in shock wave experiments [14,25,26] that could limit the signal rise time. The main goal of the study is to cross-link shock physic and Electromagnetism domain in order to improve the bandwidth of the piezoresistive pressure measurement.

In this study, we defined and addressed the rise time issues, noise disturbances and signal oscillations and related those issues with the limited bandwidth of the measurement chain, the signal noise due to electromagnetic coupling (CEM) and the capacitive effect of the gauge.

The shock gauge calibration experimental setups are presented first. The modeling of the measurement chain is then detailed for high-impedance gauges. Finally, the comparison between experiments and numerical simulations is discussed.

## 2. Experimental Setups

### 2.1. Shock Calibration Experimental Setup

The classical calibration setup uses a symmetrical impact with copper or PMMA (PolyMethylMethAcrylate) as presented in Figure 1 and Figure 2. For our copper calibration setup (Figure 1), the flyer diameter was 80 mm, and the thickness was 10 mm. The impact velocities varied between 400 and 1000 m/s to explore a 7 to 20 GPa pressure range. The transfer plate was 110 mm in diameter and 4 mm thick. A copper disk was added behind, and it was 74 mm in diameter and 10 mm thick. A high-impedance or a low-impedance piezoresistive gauge was placed in between.

Our PMMA calibration setup (Figure 2) was similar to that of [14]. The PMMA impactor had a diameter of 90 mm and a thickness of 10 mm. The PMMA transfer plate was 110 mm in diameter and 15 mm thick to avoid any deformation with the gun vacuum and while the projectile was near the impact. A first 5 mm thick PMMA disk (80 mm in diameter) was added behind the transfer plate. Two 10 mm thick PMMA half disks with the same diameter were placed after the transfer plate, with a 20 mm thick half disk in order to integrate the 25 Ohm Vishay transverse (T) gauge vertically (number 3 on Figure 2). The two 48 Ohm Vishay longitudinal (L) gauges (number 2 on Figure 2) were between the 5 mm thick disk and the first 10 mm thick half disk and the second one. The longitudinal gauges J1 and J2 and the transverse gauge J3 are illustrated in Figure 3. The impact velocities were between 300 and 1100 m/s to explore a 0.3 to 2 GPa pressure range.

The experimental setup was placed on the muzzle of a gas gun, and the gauges were electrically connected to a conditioning device called Somelec, as reported in Figure 4. Such a device integrated the reference resistance of a Wheatstone bridge (R_REFERENCE_), into a quarter configuration, as well as a three-resistors (Ro, P_1_) potentiometric system (between A and B) in order to balance the bridge, as reported in Figure 5. This potentiometric system was set by a controller to balance the Wheatstone bridge before the experiment. The Wheatstone bridge was powered by a current source (Io) shunted by a capacitor (Co). After balancing, any variation of R_GAUGE_ was translated into a variation of the bridge’s output voltage (V_DIFF_).

Six measurements channels were integrated in this Somelec device. For safety reasons, a distance in the range of five meters separated the conditioning device and the gauge. The conditioning device was connected to the gauge with a coaxial transmission line (TL).

### 2.2. Shock Calibration Experimental Results

An example of the output voltage of the Wheatstone bridge (V_DIFF_ in the Figure 5) is presented in Figure 6 for the Vishay longitudinal J1 and J2 gauges and transverse J3 gauge with a PMMA velocity impact 305 m/s. The duration of the signal is quite less than 8 µs for J1, 2 µs for J2 and more than 8 µs for J3, before the gauges brake. The digitizer bandwidth was usually 300 MHz, and the sampling rate was 2.5 Gsamples/s. The shock arrived on the gauge J1 just 75.1 µs after the start of measurement recording, 75.8 µs for J3 and 78.4 µs for J2. The analysis of the plateau was associated with the sustained shock impact condition. There were two rise slopes before and after the plateau for J1 that were often seen. The signal decreased for J3. There was also a slight rise slope for J2. These phenomena could be associated with a gauge heating effect.

### 2.3. Electromagnetic Coupling Monitoring Additional Setup

An electromagnetic disturbance source could be generated by different phenomena, such as a triboelectric effect induced by the friction of the polyethylene projectile along the steel tube of the powder or gas gun and an ionization effect due to compression of the residual air between the projectile and the target just before impact. The presence of such disturbing sources, which were unpredictable until now, was checked by measurements. Therefore, during the experiment, a wire conductor was connected between the transfer plate and the ground conductor of the experimental building electrical supply. Such a configuration enabled us to measure a potential discharging current during the mechanical impact. By inserting a Rogowski coil around this wire, as reported in Figure 7, we measured this current. A Rogowski coil was also added around the coaxial line connecting the gauge to the Somelec device in order to measure the presence of a perturbation current propagating from the gauge toward the Somelec device.

### 2.4. Electromagnetic Coupling Experimental Characterization Results

The recorded data during this particular experiment are reported in Figure 8, which depicts on a common time scale the ground-cable current detected by the Rogowski coil (black curve/right scale), the perturbation current propagating from the gauge toward the Somelec device (blue curve/right scale) by another Rogowski coil and the output voltage measured on the Wheatstone bridge (red curve/left scale). A correlation can be made between the output voltage and the discharging currents (to the ground as well as toward the Somelec device). A few microseconds before the shock on the gauge, which was around the time 267 µs, this correlation was very clear. Before this time, the absence of mechanical stress on the gauge enabled us to estimate the transfer impedance between the output voltage and this disturbing current. Such analysis is reported in the next sections of this paper. It is, however, evidence that there is an electrical charge injection near the mechanical impact time. Furthermore, the estimation of the transfer impedance is compliant with the observed data after impact (after 267 µs). The visible disturbance on the output voltage after the impact is almost explainable by a transfer effect modeled by our transfer impedance estimation. In a conventional experimental setup (without a wire connection of a transfer plate to the ground conductor), this perturbation current can propagate entirely inside the coaxial line by means of the capacitive coupling of the gauge, no matter the quality of the experimental setup’s electrical shielding. The usage of Rogowski coil around the coaxial line during an experiment confirmed this phenomenon. The transmission line, which connected the gauge to the Somelec device, transmitted the measurement signal according to a differential propagation mode. The perturbation current propagated inside the transmission line according a differential mode as well as a common mode. The common mode was detected by the Rogowski coil. As explained in the next sections of this paper, the architecture of the Somelec device was sensitive to these two kinds of disturbances. The part of the perturbation current, which propagated according to the differential mode, introduced an electrical noise on the measurement signal. The other part of the perturbation current, which propagated according to the common mode, was transferred on the output voltage due to the architecture of the Somelec device.

In order to understand the unpredictability of the disturbance presence during an experiment, we investigated the coupling origin between the gauge and the transfer plate. A complete modeling of this experimental setup enabled us to analyze and predict the time response of the measurement system as well as the theoretical accuracy of the measurement. This modeling started with the analysis of the interactions between the stress sensor, the gauge and the physical environment around it. The electromagnetism theory explains these interactions. Such investigations are reported in the following section of this paper.

### 2.5. Origin of the Capacitive Coupling between the Gauge and the Transfer Plates

The measurement system’s rise time was related to the influence of the shock reflections and equilibrium within the gauge and was proportional to the gauge thickness. The general assumption was the following: 50 µm gauge thickness—50 ns, 100 µm—100 ns and 150 µm—150 ns [4], which could be more accurately estimated for each experiment by a hydrocode, considering the glue and impedance-matching materials. The gauge was electrically connected to the other remote components with a TL to constitute a Wheatstone bridge in quarter-bridge configuration. In order to avoid any current leakage onto the transfer plate, the gauge was embedded into a dielectric sheet. According to the transfer plate chosen for the experiment, a conductive material could be located close to the gauge. This configuration implied a capacitive leakage between the gauge legs and the transfer plate. This leakage can be modeled by a capacitance [27], which can be estimated using the well-known Equation (2).
(2)$$C=\frac{\varepsilon \cdot A}{d}$$
where *d* is the distance between the transfer plate (m) and the gauge, *A* is the area of gauge (m^2^) and its legs and *ε* is the dielectric constant. In the case of a conductive transfer plate, the value of this capacitance is lower but can be estimated by calculating the product of the linear capacitance *Co* between the gauge’s legs over the gauge’s legs’ length *l*. This can be performed analytically or by using any transmission line code. In the same way, it is possible to estimate the linear inductance *Lo* between the gauge’s legs over *l* in order to model the effect of the electromagnetic leakage between the gauge and the transfer plate by a lumped reactive elements circuit on the gauge’s output. Such analysis of capacitive leakage has been performed [28] but the effect of the impactor on *C* or *Co* when it reaches the target has not yet been reported. It is reasonable to expect that the mechanical deformation of gauge versus time during the impact modulates the instantaneous value of *C* or *Co* and affects the time domain shape of the measurement signal. This impactor effect modeling was not performed but is targeted for a next publication. The capacitive coupling phenomenon between the transfer plate and the gauge has been implemented into a multi-physic model of gauges in order to be able to describe the right time domain response of the stress sensor during a shock experiment. This modeling is reported in the next section of this paper.

## 3. Modeling of the Measurement Setup

Based on the capacitive coupling characterization, the modeling of the measurement chain takes into account the following steps: the gauge itself and the related bandwidth limitation, the transmission line influence, the conditioning system transfer impedance and the charge induction.

### 3.1. Gauge Modeling

The gauge’s behavior during the target impact can be modeled by a time-varying resistance *R*(*t*) controlled by the applied stress *σ*(*t*) versus time on the gauge. Using expression (1), the time variation of the gauge resistance *ΔR*(*t*) can be expressed by Equation (3) if dilatation strain remains constant over time.
(3)$$\Delta R(t)=R_{0}\cdot \Delta \sigma (t)\cdot a$$

Such an equation can be implemented into an electrical simulator such as Advanced Design System (ADS). Thus, this multi-physic model was integrated into ADS using the following analogy: one unit of stress (*σ*) is equivalent to one unit of a pseudo-voltage source (*V_DRIVE_*), which drives the time variation of a voltage-controlled resistance. Therefore, the following Equation (4) was implemented on ADS.
(4)$$\Delta R(t)=R_{0}\cdot \Delta V_{DRIVE}(t)\cdot a$$

A gauge’s model was then used within the ADS simulator to perform an electrical simulation of the measurement channel for a pressure measurement setup in the time or frequency domain. This model was completed by the electromagnetic effects that were located near the gauge’s legs as well as the electrical disturbance sensitivity of the measurement setup. A capacitance *Cp* modeled, in the electrical domain, the effect of the transfer plate’s presence on the gauge’s bandwidth. The capacitive leakage between the gauge and the transfer plate generated a displacement current that propagated between the gauge legs via the transfer plate. Combined with its internal impedance (*Ro* for a small signal modeling), the gauge behaved as a low-pass filter. The other consequence of having a transfer plate near the gauge was to induce electrical charges from the impactor toward the gauge. This phenomenon was modeled by the capacitance *Cs* that were located between the disturbance source (the electrical charge source) and the gauge. Each of these capacitances (*Cp*, *Cs*) were related to the gauge sample as well as the transfer plate’s material and its geometrical dimensions. The description of our gauge model is depicted in Figure 9.

This gauge model was used to design a simulation on ADS of our measurement benchmark. This simulation setup included the controlled current source of the Somelec device, the reference resistor, the potentiometric bridge, the load impedance on the output channel of the Wheatstone bridge, which was modeled by the digitizer input impedance, and the coaxial cable used to connect the digitizer to the Somelec device. The coaxial TL used to connect the gauge to the Somelec device was also integrated into the simulation. Our simulation tool enabled us to analyze the effects of the wave’s propagation on the time response or bandwidth of the measurement channel. The next paragraph explains the effect on the bandwidth.

### 3.2. Bandwidth Limitation

The gauge modeling enabled us to design a response model of a measurement bench in order to identify the phenomena that originated from the observed unpredictable disturbances as well as the mismatch between the theoretical response’s time delay of the gauge and the recorded transients on the measurement signal. In order to do this, each element of the bench must be modeled. Thus, an electrical model (small signal modeling) of the apparent impedance exhibited by the Somelec device was performed. This model is summarized as the reference resistor *Ro* integrated into the Somelec device with its parasitic reactance (inductance *Lo*). Because the gauge was connected to the Somelec device with a coaxial line, an electrical model of the coaxial TL was then included in our simulation. The well-known rule of propagation waves theory [29] shows that the presence of a TL in an electrical circuit cannot be neglected if its length is higher than a tenth of the signal wave length. This was the case in our application, where the gauge was separated from the Somelec device by at least five meters.

The bandwidth limitation of the measurement channel is explained by the low-pass frequency effect of electrical circuit depicted in Figure 9. The time response of such a filter was related to the well-known *Ro*Co* product. According to the chosen materials of the transfer plate (*Co*) and the gauge used during the experiment (*Ro*), this parameter dominated and controlled the global time response of the measurement channel. In other cases, this was the standing wave ratio (SWR) in the TL, which explained the bandwidth limitation of the measurement channel. The reference impedance of the Wheatstone bridge, which was connected to the end of the coaxial TL, significantly deviated from a purely real impedance *Ro* (matched to the characteristic impedance of TL) toward a complex impedance (5). This resistance exhibited a series parasitic inductance *Lo*. The internal impedance of the Somelec device input (*Zin*) was then given by the following complex expression (5):*Zin* = *Ro* + *j***ω***Lo*(5)

Such a phenomenon degrades the level of power return loss at the end of the coaxial TL and thus the SWR. This property is responsible for the resonant behavior of the frequency response of the measurement channel in the range of high frequency (above 10 MHz). All these limiting parameters can partly be offset by a frequency compensation process if the signal-to-noise ratio (SNR) in the range of the cut-off frequency remains low. However, such a signal processing is unable to overcome the frequency behavior of the measurement setup. This was the case with our experimental setup, where an unpredictable and important electrical noise signal was present on the measurement signal. This behavior did not enable any frequency compensation process nor did it accurately measure the pressure step levels around a mechanical impact. This topic is addressed in the following section.

### 3.3. Cable Influence and Conditioning Transfer Impedance

All components used to transmit a measurement signal are passive (resistors of Wheatstone bridge, cables and shielded cables). Thus, if an abnormal electrical noise is present in the measurement signal, the origin of this noise is located outside the measurement bench and is transferred into it. If a transient electrical disturbance is induced onto the transfer plate, this disturbance can propagate to the gauge legs by the capacitive coupling. This is the consequence of the capacitive leakage between the transfer plate and the gauge as reported in the previous sections of this paper. The induced electrical disturbance can be modeled by a time-varying current source *Ip*(*t*) connected between the ground potential and the transfer plate. Thus, the current source *Ip*(*t*) used in our behavioral model represented the injection of electrical charges from the impactor toward the gauge. The magnitude and the time-varying shape of *Ip*(*t*) were extracted from the historical measurements performed in the previous experiments with Rogowski coils. *Ip*(*t*) was very low, and no significant disturbance was detected before or after the impact. However, during a time period around the target impact, a significant current impulse was detected, as reported in Figure 8.

Finally, the schematic of the equivalent electrical model now includes the effects of the presence of the transfer plate and the effect of the electrical disturbance that is generated on the measurement bench in its time domain response. This schematic is depicted in Figure 10, as well as the travel of a disturbance current along the TL and its effect on the measurement signal. This model was validated step by step: first by the arrival time of the electrical wave, and then by the TL transfer impedance. The reported schematic in Figure 10 shows that the disturbance current propagated along the TL according to a differential mode and a common mode. The magnitude ratio between these two modes was unknown because the Rogowski coil was not appropriate for measuring a differential mode current on a coaxial cable. The repartition of the disturbance current into the Wheatstone bridge under these two propagation modes explains its transfer on the output voltage of the bridge V_D_ despite the symmetry of the bridge.

Each of the following effects, capacitive coupling between the transfer plates and the gauge (*Cp*), the cables’ influence (TL), as well as the Somelec input impedance (*Zin = Ro + j.Lo.ω*), were depicted by our bench measurement model. According to the extracted component models performed during our investigations, we plotted the normalized gain of bench measurement versus frequency. The result of this simulation is reported in Figure 11.

We can see the bandwidth limitation effect related to the impedance mismatch in the range of high frequencies (upper to a fewer megahertz). Unfortunately, we have not yet measured this measurement bench characteristic. Such a measurement requires a specific experimental setup that we must design. This task is targeted in the future in order to correctly calibrate the measurement setup in the dynamic domain.

The last effect depicted by our bench model was the electrical charge induction from the impactor to the gauge (*Cs*). This point is reported in the next section of this paper.

## 4. Comparison between Experiments and Numerical Simulations

The charge induction was reproduced onto the transfer plate using a specific setup. In order to do this, we injected electrical charges onto the transfer plate using a sinusoidal current source connected to the transfer plate. A Rogowski coil was inserted around the coaxial line in order to measure the injected current’s magnitude. The voltage provided at the Wheatstone bridge output was measured by a digitizer. We also detected the presence of a significant disturbance voltage on the measurement channel due to the input of a digitizer. The ratio between the voltage and current disturbances led to a transfer impedance in the range of 10–25 Ohms, which was not compliant with a typical shielding defect in the TL setup. The electrical configuration of the Somelec bridge and the ground connection of the chassis (or digitizer) provided a local dissymmetry that explained the transfer of this current, which propagated to the output voltage of the Wheatstone bridge according a differential mode and a common mode into the TL. The named “Gauge” input channel of the Somelec chassis is sensitive to the common mode current. This sensitivity was estimated during a shock experiment (see Figure 8 and previous sections).

In order to evaluate the accuracy of our measurement bench model, we compared the measured level of the measurement chain transfer impedance to the theoretical one given by our theoretical model. The ratio between the voltage magnitude at the bridge output and the injected perturbation current magnitude is then calculated. The measured transfer impedance is plotted in Figure 12 (blue curve).

The theoretical result provided by our behavioral model was also reported in the same plot with the red curve and reproduced the general behavior of the experimental one. The gap between the two curves was compatible with a mean relative error measurement, estimated in the range of 20%, due to the impossibility to accurately know the ratio between the differential and common propagation modes as well as the unidentified electrical behavior into the bench device that was not modeled. The observed magnitude of the measurement chain transfer impedance was very high and not compliant with a shielding effect. The model validation shown in Figure 12 demonstrates that the presence of electrical noise on the measurement signal can be explained by the modeled capacitive leakage between the impactor, the transfer plates and the gauge.

## 5. Conclusions

In this study, we identified a capacitive effect of the piezoresistive gauge in the shock physic impact configuration. This capacitive effect has an important influence on the limited bandwidth of the measurement chain with the Somelec Wheatstone bridge and on the signal noise due to electromagnetic coupling (CEM). During the experiments, disturbance currents in the range of 30 mA were observed. Such values are compatible with the presence of a parasitic capacitance in the range of several tens of picofarads and voltage variations in the range of several volts over a few microseconds. Such a current is too high for the Somelec Wheatstone bridge device. Gauge calibration shock experiments were performed with Rogowski coils on the measurement cables and on the ground cables to understand the sources of the signal noise. The current parasites on the cables were quantified depending on the configuration. The gauge and the measurement chain were modeled using ADS code, and the calculations reproduced the time arrival of the electrical wave, the transmission line transfer impedance and the conditioning system transfer impedance. The bandwidth limitation was then predicted.

The bandwidth limitation of the measurement channel was explained by the low-pass filter effect of the gauge’s internal resistance *Ro* and its parasitic capacitance *Co*, which depends on the transfer plate material. This was the main characteristic that limited the bandwidth capability of the experimental setup. Another source of limitation was the mismatch between the Somelec device’s input internal impedance and the transmission line. Such behaviors forbid the observation of very short transient’s phenomena. Finally, the Somelec device was sensitive to the injection of electrical charges during measurement. The observed magnitude of the measurement chain’s transfer impedance did not comply with a shielding effect. As this disturbance sensitivity is intrinsic to the Somelec device, if an electrical discharge is injected inside the coaxial cable via the pressure sensor, an accurate pressure measurement is impossible.

The investigation is to determine whether a new piezoresistive pressure gauge system is possible with the manganin pressure gauge and if it could avoid the transfer of the electrical disturbance on the measurement signal. Work is ongoing to design a new measurement setup with a balanced structure.

## Figures and Tables

**Figure 1 sensors-23-06583-f001:**
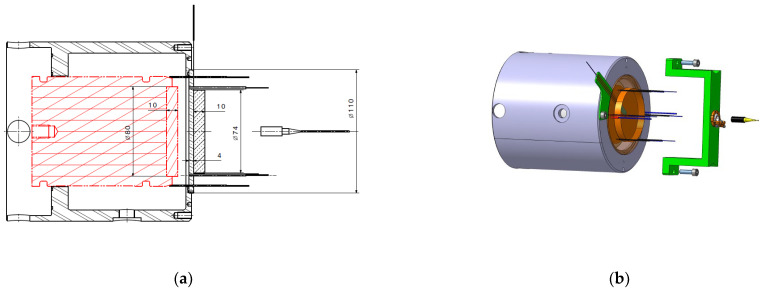
Detail of the copper calibration experimental setup. (**a**) Side view. (**b**) 3D view (quotations are in mm).

**Figure 2 sensors-23-06583-f002:**
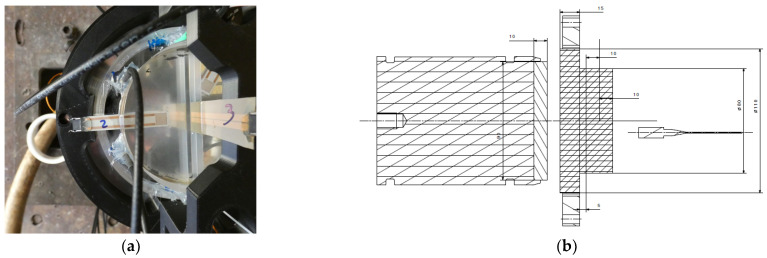
Detail of the PMMA calibration experimental setup. (**a**) Photo of the experiment with the three gauges J1, J2 and J3. (**b**) Side view (quotations are in mm).

**Figure 3 sensors-23-06583-f003:**
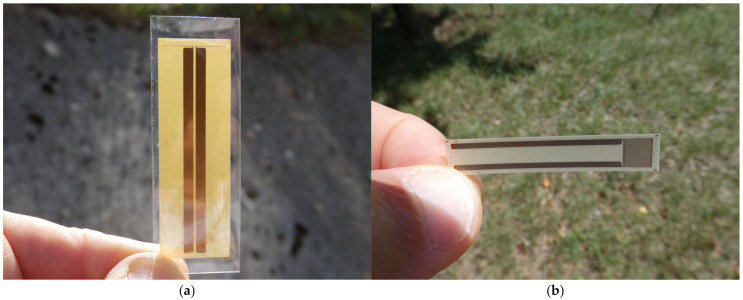
Examples of (**a**) 25 Ω Vishay T gauge and (**b**) 48 Ω Vishay longitudinal gauge.

**Figure 4 sensors-23-06583-f004:**
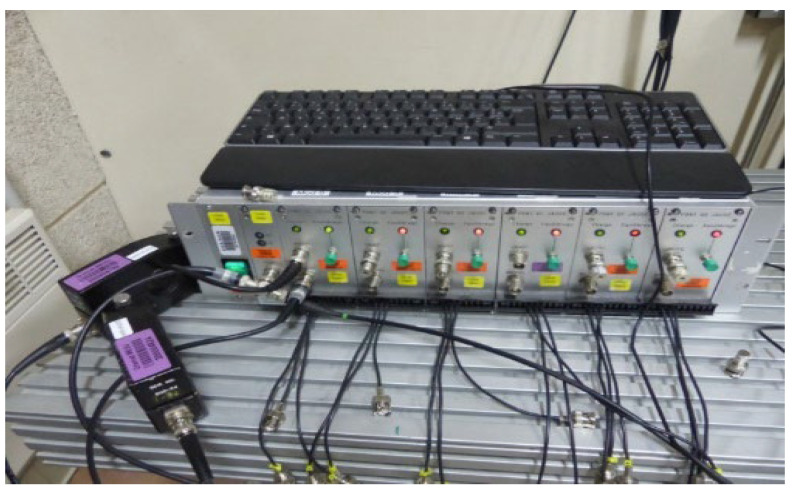
Somelec Wheatstone bridge and calibrated Rogowski coils on different cables.

**Figure 5 sensors-23-06583-f005:**
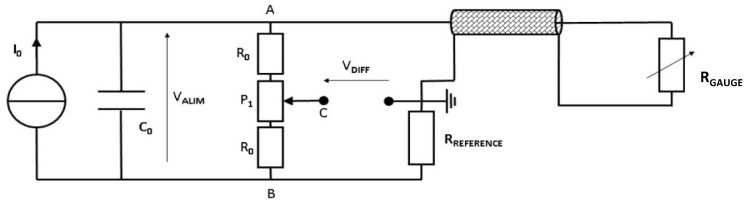
Equivalent electrical schematic of the Somelec Wheatstone bridge. Io: current source; Co: capacitor; first side with Rgauge and Rreference and second side Ro and P_1_; potentiometric system resistors between A and B; Valim: power supply tension; and Vdiff: output tension.

**Figure 6 sensors-23-06583-f006:**
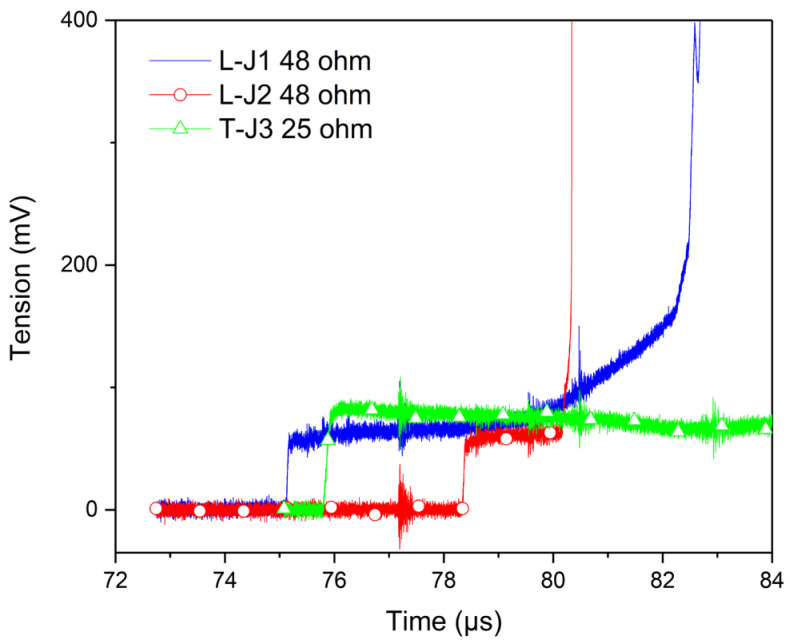
Example of output voltage on the longitudinal gauge L-J1, longitudinal gauge L-J2 and transverse gauge T-J3 for an impact velocity of 305 m/s.

**Figure 7 sensors-23-06583-f007:**
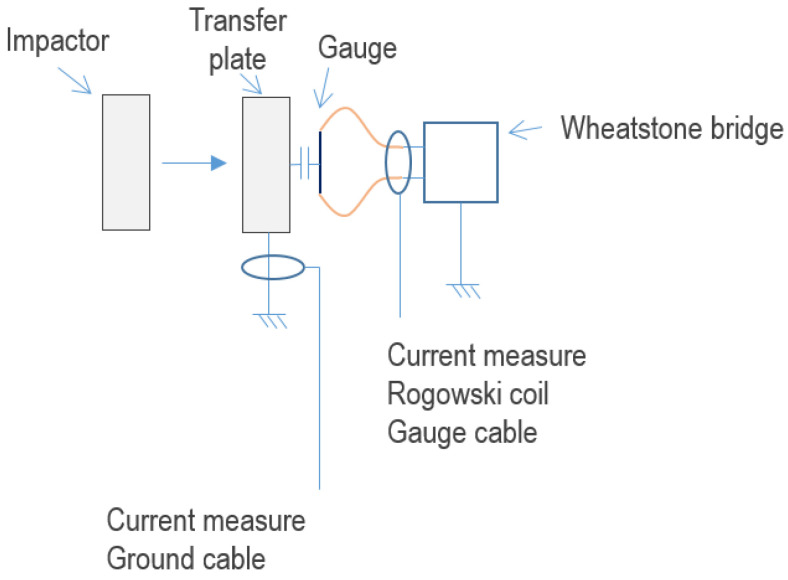
Scheme of the electromagnetic coupling setup with a Rogowski coil on the measurement cable and on the additional transfer plate ground cable.

**Figure 8 sensors-23-06583-f008:**
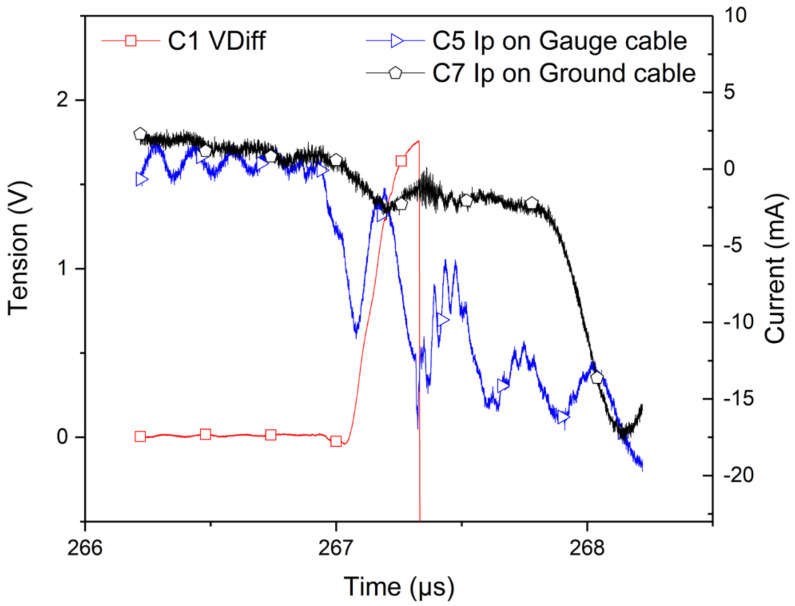
Current signals (with Rogowski coil) on the coaxial line connecting the gauge and the Somelec device (blue curve/right scale) inside a wire connecting the copper transfer plate (black curve/right scale) and the ground, compared with the output voltage (red curve/left scale) recorded with a longitudinal gauge for a copper/copper configuration with an impact velocity of 403 m/s (at time 267 µs on the gauge).

**Figure 9 sensors-23-06583-f009:**
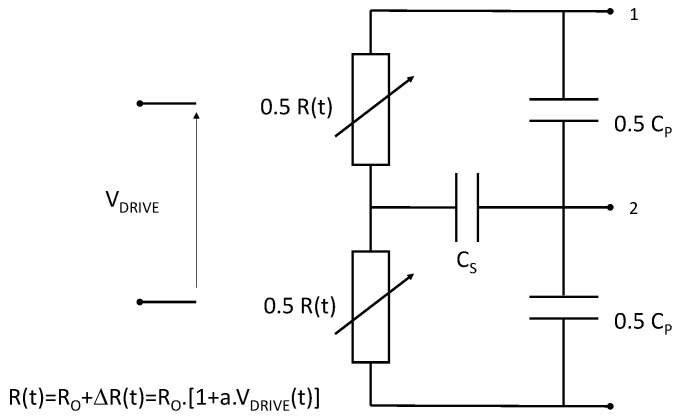
Detail of the gauge model implemented in ADS code.

**Figure 10 sensors-23-06583-f010:**
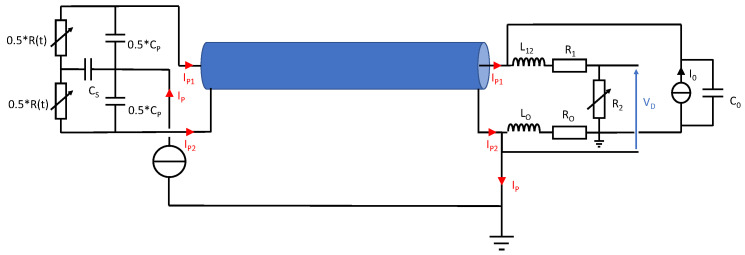
Schematic of the equivalent electrical model and the electrical disturbance injection passing through the modeled measurement chain.

**Figure 11 sensors-23-06583-f011:**
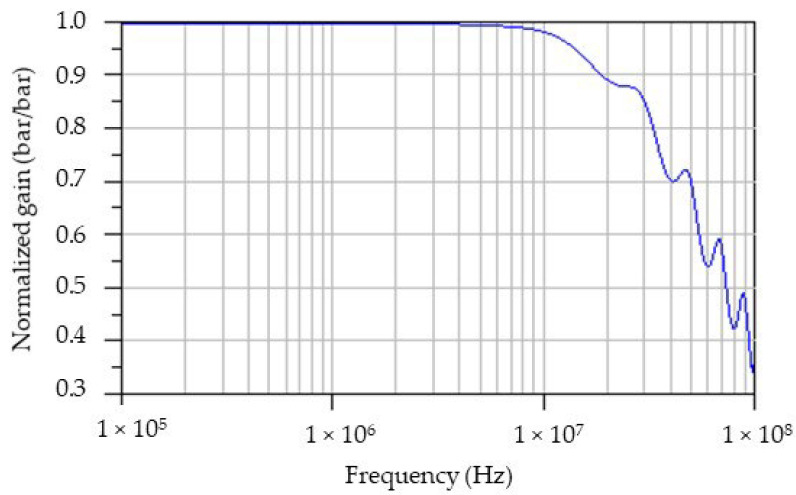
Normalized gain (in bar/bar) of measurement bench versus frequency.

**Figure 12 sensors-23-06583-f012:**
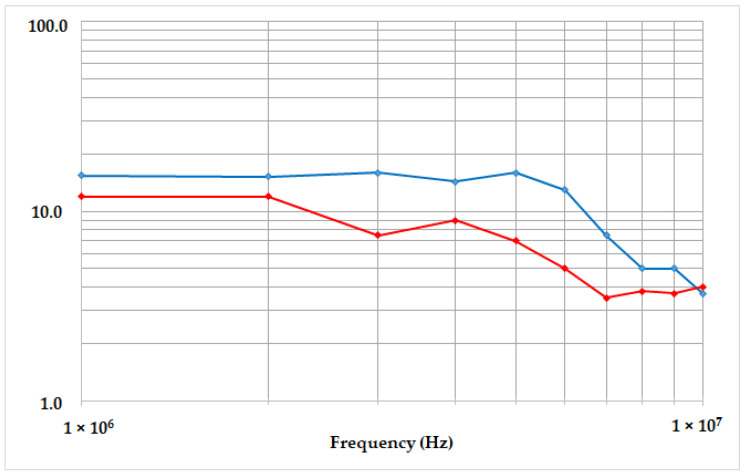
Measurement chain transfer impedance versus frequency, measured transfer impedance (blue curve) and the theoretical one provided by our modeling (red curve).

## Data Availability

The data presented in this study are available on request from the corresponding author.

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
