# Peer review of "Capacitive Effect and Electromagnetic Coupling on Manganin Gauge Limiting the Bandwidth for Pressure Measurements under Shock Conditions"

_sensors, 2023, doi:10.3390/s23146583_

Round 1

Reviewer 1 Report

This is a very interesting paper on a high quality work. The numerous physics insights show that the authors have strong command on their subject. Simulations are in accordance with observations. 

Minor polishing would be a good idea.

Author Response

The authors would like to thank you very much for your comments and suggestions.

The article was read and cross read by members of the teams, who are not coauthors, to improve the English language.

Reviewer 2 Report

Thanks to the authors for their valuable work with a fantastic modelling of the measurement setup, but a minimum revision of the structure and content of the article is needed:

·      The introduction is very short. It is necessary to complete the state of the art by increasing the bibliographical references if possible.

·      Differences between the transfer impedance of the measurement chain and the theoretical one provided by the modelling must be explained in terms of the measured frequency ranges, line 351, even if a reliable statistical fit can be obtained with the 2 curves, indicating the average relative error and the goodness of fit with the correlation coefficient.

·      The conclusions are very general. It is necessary to quantify with data and values such as the values of eddy currents in each of the configurations, line 370.

·      The investigation is to determine whether the new piezoresistive pressure gauge system is possible with the manganin pressure gauge and if it could avoid the transfer on the electrical disturbance on the measurement signal, line 386.

Author Response

The authors appreciate yours comments, suggestions and questions on the paper, it helps to improve the article. We would like to thank you very much sincerely.

The introduction is developed as proposed and allow to identify the main research areas: gauge calibration, shock and release response, gauge elasto-plastic behavior influence, noise disturbance issues and conditioning devices. Here are some additional references to complete the state of the art:

Morris, C., E., Los Alamos shock wave profile data, 1982; University of California Press, ISBN-0-520-04007-4

Armstrong, C., Rae, P., Tasker, D., Heatwole, E., Broilo, B., Inexpensive method of qualifying piezoresistive thin-film pres-sure gauges, Shock Compression of Condensed Matter, St-Louis, Missouri, 2017; LA-UR-27852.

Chapman, D.J., Braithwaite, Proud, W.G., Calibration of wire-like manganin gauges for use in planar shock-wave experi-ments, Shock Compression of Condensed Matter, Nashville, Tennessee, 2009; 603-606 ; AIP Conference Proceedings 1195, 603.

Bernstein, D., Godfrey, C., Klein, A., Shimmin, Research on manganin pressure transducers, Symposium High Dynamic Pressure, Paris, 11-15 September, 1967.

Bourne, N., K., Rosenberg, Z., Fractoemission and its effects upon noise in gauges placed near ceramic interfaces, Shock Compression of Condensed Matter, Seattle, Washington, 13–18 Aug, 1996; American Institut of Physics conference proceed-ings 370; 1053-1056.

The differences between the transfer impedances are explained as proposed. The gap between the two curves is compatible with a mean relative error measurement estimated to the range of 20%, due to the impossibility to known accurately the ratio between the differential and common propagation modes as well as the unidentified electrical behavior into the bench device, which are not modelled. The curves are just point by point drawing. A fit has not been proposed for each data set in order to assess the shift directly from the raw data.

Quantitative data have been added to the conclusion. During experiments, disturbance currents in the range of 30 mA have been observed. Such values are compatible with the presence of a parasitic capacitance in the range of several tens of picofarads and voltages variation in the range of several volts over a few micro-seconds. Such a current is too high for the Somelec Wheatstone bride device.

The end of the conclusion has been clarified as proposed.

Reviewer 3 Report

The paper analyses the capacitive effect and the electromagnetic coupling on the measurement chain induced by the gas gun or the powder gun during impact experiments.

R1. The introduction should be extended to include other significant results and a clear explanation of the contribution of this paper. In its current form, there are too many references for just one effect [1-9] and a very brief paragraph about the paper's main goal.

R2. Chapter 2 should have a clear delimitation between the experimental setup and the performed tests. The first part should focus only on the setup, while the other subchapters should detail the tests performed.

R3. Chapter 3 should have very clear traceability from the test performed to the model developed. At present, it is a little difficult to identify this connection.

R4. The discussion about the obtained results should be extended. There are very valuable results that are just named and not explained. 

R5. Figure 5 should have labels.

The quality of the English Language is fine.

Author Response

The authors appreciate the remarks on the paper, which helps to improve it a lot. Thank you very much sincerely.

R1. The introduction has been improved as proposed.

“The main research have been focused on the piezoresistive gauge calibration for low impedance gauges [13, 14, 15, 16, 17] and high impedance gauges [14, 15, 18], on the conditioning device development [6, 12, 13] and on gauge issues, such as hysteresis [14, 19], piezoresistive behavior during the release [20], piezoresistive behavior in the elastic domain, and in the plastic domain of the gauge [21, 22]. The signal rise time of the gauge, limited to its thickness, has been demonstrated [23]. Noise disturbance have been observed and mitigated previously in shock wave experiments [14, 24, 25], that could limit the signal rise time. The main goal of the paper is to cross-link shock physic and Electromagnetism domain in order to improve the bandwidth of the piezoresistive pressure measurement.”

R2. The chapter 2 takes into account new subchapter to present separately the shock experimental set-up, the electromagnetism coupling monitoring additional set-up, the shock experimental results, the Electromagnetism coupling monitoring results, and the origin of the capacitive coupling between the gauge and the transfer plates.

R3. A better link between chapter 2 and chapter 3 is proposed at the end of paragraph 2.4:

“A complete modelling of this experimental setup enables to analyze and predict the time response of the measurement system as well as the theoretical accuracy of the measurement. This modelling starts with the analysis of the interactions between the stress sensor, the gauge, and the physical environment around it. The electromagnetism theory explains these interactions.”

A link has been also added at the beginning of the chapter 3:

“Based on the capacitive coupling characterization, the modeling of the measurement chain takes into account the following steps: the gauge itself and the related bandwidth limitation, the transmission line influence, the conditioning system transfer impedance and the charge induction.”

R4. The bandwidth limitation has been more detailed at the end of chapter 3. The comparison between experiments and simulations has been extended in chapter 4 with a quantitative value, and an explanation of the discrepancy.

R5. We improve figure 5 and add the symbol explanation in the text and in the caption
